# Removal of Toxic and Essential Nutrient Elements from Commercial Rice Brands Using Different Washing and Cooking Practices: Human Health Risk Assessment

**DOI:** 10.3390/ijerph19052582

**Published:** 2022-02-23

**Authors:** Syfullah Shahriar, Alok Kumar Paul, Mohammad Mahmudur Rahman

**Affiliations:** 1Global Centre for Environmental Remediation (GCER), College of Engineering, Science and Environment, The University of Newcastle, Callaghan, NSW 2308, Australia; 2Department of Soil Science, Sher-e-Bangla Agricultural University, Dhaka 1207, Bangladesh; alokpaulsau@yahoo.com

**Keywords:** trace elements, rice, essential elements, washing and cooking, dietary intake, health risk

## Abstract

This study determined the influence of different cooking procedures on the removal of toxic elements (TEs) including arsenic (As), cadmium (Cd), and lead (Pb) along with other nutrient elements from different commercially available rice brands sold in Bangladeshi markets. We observed 33%, 35%, and 27% average removal of As, Cd, and Pb accordingly from rice when cooked with a rice to water ratio of 1:6 after washing 5 times. We also found a significant reduction in essential elements: Zn (17%), Cu (10%), Mn (22%), Se (49%), and Mo (22%), when rice cooking was performed as in traditional practice. Daily dietary intakes were found to be between 0.36 and 1.67 µg/kgbw for As, 0.06 and 1.15 µg/kgbw for Cd, and 0.04 and 0.17 µg/kgbw for Pb when rice was cooked by the rice cooker method (rice:water 1:2), while in the traditional method (rice:water 1:6) daily intake rates ranged from 0.23 to 1.3 µg/kgbw for As, 0.04 to 0.88 µg/kgbw for Cd, and 0.03 to 0.15 µg/kgbw for Pb for adults. The HQ and ILCR for As, Cd, and Pb revealed that there is a possibility of noncarcinogenic and carcinogenic risk for As but no appreciable risk for Cd and Pb from consumption of rice.

## 1. Introduction

Rice is consumed by millions of people as their carbohydrate staple, which supplies more than half of the need of caloric intake by humans throughout the world [1]. For example, the population of Bangladesh obtains 73% of their total caloric intake from rice [2]. Rice contains various micro- and macronutrients including zinc (Zn), copper (Cu), manganese (Mn), selenium (Se), and molybdenum (Mo), which make it suitable for uptake of these nutrients for the people of Asian countries who can rarely afford expensive fruits or other food items. Apart from these nutrients, rice is also considered as an exposure source to multiple toxic elements (TEs) including arsenic (As), cadmium (Cd), and lead (Pb). Both As and Cd are toxic elements, recognized as Class I carcinogens. The bioaccumulation of As in rice has been a topic of worldwide resonance in the last thirty years, and many efforts have been made to reduce its bioaccumulation in the edible fraction, even in heavily polluted soils including water management options [3,4,5,6,7,8,9,10]. The dietary intake of As and Cd and the health risk to people consuming contaminated rice have been reported globally [11,12,13,14,15,16,17]. People from the As-affected countries are considerably exposed to As via the consumption of food, especially rice [15,16,17]. Concentration of As may differ between raw rice and cooked rice and depends on the rice cooking process [18,19,20]. Rice is usually washed with water until clear and, finally, cooked traditionally with excess water. After cooking, the excess water is discarded [21]. The concentration of As in cooked rice usually varies based on the content of As in the raw rice, the As in the cooking water, and the cooking procedures [22,23,24]. A study reported that up to 57% of As can be removed when rice is washed until clear, then cooked with 1:6 rice and water, and excess water is discarded after cooking [21]. Another study reported that cooking rice with deionized water (1:6 ratio) can remove around 35% and 45% of total and inorganic As concentrations from long-grain and basmati rice, respectively [25]. Similarly, cooking of rice with excess water (1:10 rice: water ratio) removed inorganic As by 50%, 60%, and 40% in brown rice, parboiled, and long grain polished rice, respectively [26].

Food is the main source of exposure to Cd for the non-occupationally exposed population. Among different food items, rice alone accounts for more than 30% of the total dietary intake of Cd [27]. So, assessment of the concentration of Cd in different states of cooked rice and its associated risk to human health is of increasing concern. Though dietary intake of Cd from different countries was reported to be safe [28,29,30,31,32,33], there are several other studies, which reported a health risk for people consuming Cd-contaminated rice regularly [11,12,13,14]. Human health risk assessment for lead (Pb) and other toxic metals from rice consumption has also been reported in various countries [32,34,35,36].

The quality of cooked rice is largely dependent on the various factors involved in rice preparation. Hence, the rice to water ratio is an important issue [37]. In Southeast Asia, the traditional method of rice cooking practiced involves washing of the rice three to five times (until the water is clear), followed by cooking in excess water (four to six times the amount of rice) until an edible texture is gained; later, excess water is discarded [21]. The influence of cooking rice with excessive water (1:6 to 1:12 rice to water ratio) has been investigated in different studies. Atiaga, et al. [38] reported that cooking of white and brown rice with excess water (1:6 rice: water) can reduce inorganic As by 60% (range 29–90%). Similarly, a 40% and 50% reduction in inorganic As from long white and brown rice accordingly was reported by Gray, et al. [26], when rice grains were cooked with a high rice to water ratio (1:6–10). Carey, et al. [39] also found a 53% reduction in inorganic As from white rice when cooked with a larger amount of water (1:12 rice to water ratio).

Another rice cooking method known as the absorption method (rice cooker) is also popular in western countries [21], in which rice cooking is performed in a rice cooker with lower rice to water ratio (1:2 to 1:3), until the water is absorbed. This method was found to be effective in retaining the essential nutrients in rice as no water is discarded, but removal of As was minimal compared to the traditional method [38,40]. It requires less time and water and was recommended to cook fortified rice to prevent the added nutrients loss.

Very few studies have investigated the concentration of TEs and micronutrients in cooked rice [1,18,26]. The major concern was As removal from rice by various cooking procedures, but data on As speciation (inorganic and organic), Cd, and Pb, along with loss of essential nutrient elements are still lacking with regard to the suitable rice washing and cooking method for different rice grains. In this study, we focused on different washing methods along with cooking procedures using various rice to water ratios (1:2, 1:3, 1:4, and 1:6) to figure out the most efficient TEs removal method. This study also evaluated the removal of essential nutrients from rice by different washing and cooking methods. We also performed risk estimation based on inorganic As, Cd, and Pb present in the cooked rice to report the actual health risk to the consumers. Based on the above, the main objectives of this study were to

(a)Measure the concentration of toxic elements (TEs) and some essential elements in rice brands available in Bangladeshi markets;(b)Investigate the effect of different washing and cooking procedures to minimize the TEs (including inorganic As) in rice and to find the most efficient method for removing TEs from rice; and(c)Assess the actual health risk based on the consumption of cooked rice.

## 2. Materials and Methods

### 2.1. Sample Collection

Ten widely consumed brands of rice (Appendix A) were purchased from Dhaka, Bangladesh during December 2019. We randomly choose one of the largest rice trading markets (Mohammadpur) of Dhaka city to collect the samples. The majority of the commercial rice varieties available in Bangladesh is parboiled, as it is the common practice of rice grain preparation in Bangladesh, and all the rice varieties used in this experiment were parboiled.

### 2.2. Washing of Rice

Raw rice (100 g) was placed in a Pyrex beaker and washed 3 and 5 times with 300 mL of tap water (concentrations of As, Cd and Pb were below detection limits), and all 3- or 5-times washed water samples were combined in a 1 L prewashed plastic container. Then, 10 mL of the water sample was taken for further analysis to determine the reduction in TEs due to washing. After washing, the raw rice was allowed to dry at 60 °C for 48 h. Different ratios of water were then added with the rice to conduct the cooking experiments.

### 2.3. Cooking of Rice

We used two common methods of rice cooking in our study. (1) The rice cooker method: The washed rice was cooked in a rice cooker with 200 mL and 300 mL of double-distilled deionized water (rice to water ratios 1:2 and 1:3) until no water was left to discard. (2) The traditional method: this method is still in use by more than 90% of the villagers in the Bengal delta regions of Bangladesh and India. The washed rice was subjected to 1:4 and 1:6 rice: water cooking in a glass beaker until the texture of eating; then, the remaining water (gruel) was discarded. Appendix A shows the flow chart of the washing and cooking procedures conducted in this study. All experiments were performed in triplicate and data were reported as mean ± standard deviation (SD).

### 2.4. Sample Processing, Digestion, and Analysis

All raw rice and cooked rice samples were oven dried at 65 °C for up to 72 h and then crushed using a stainless-steel crusher. All powdered samples were stored in ziplock bags and kept in a desiccator. Ground rice grain samples were digested by the heating block digestion method according to Rahman et al. [15]. The detailed digestion procedure is given in Appendix A. The digested samples were analyzed by the inductively coupled plasma mass spectrometry (ICP-MS, 7900, Agilent Technologies, Hachioji-shi, Japan) for TEs and nutrient elements. The concentrations of TEs and nutrient elements in rice samples were expressed on a dry weight (dw) basis. The open block digestion procedures for measuring As, Cd, and Pb in food samples were validated and reported previously [16,28,41]. The instrumental lowest detection limits (LODs) for As, Cd and Pb were 0.01, 0.03, and 0.05 µg/L, respectively [18,42]. The instrumental lowest quantification limits (LOQs) for As, Cd, and Pb were 0.03, 0.09, and 0.15 µg/L, respectively [18,42].

Speciation analysis of As, including inorganic As (arsenite, As(III), and arsenate, As(V)), monomethylarsonic acid (MMA), and dimethylarsinic acid (DMA) was conducted according to the procedure of Signes-Pastor, et al. [43] using 1% nitric acid in a microwave oven and analyzed by the IC-ICP-MS. The detailed procedure was described in Islam, et al. [44]. In this study, the concentration of inorganic As (sum of As(III) and As(V)) was reported. The speciation method of As was validated and reported elsewhere with LODs [41,44,45,46,47].

### 2.5. Estimation of Daily Dietary Intake of As, Cd, and Pb from Rice

Based on the consumption of rice and the mean concentration of TEs in rice (dry weight) observed in this study, we estimated the daily estimated dietary intake (EDI) of As, Cd, and Pb from rice by adults, using the following equation:$$\mathrm{EDI}=\frac{\mathrm{CRa}\;\times \;\mathrm{CEa}}{\mathrm{BW}}$$
where

EDI = Estimated daily consumption (µg/kg bw/day);

CRa = Intake rate of rice (0.432 kg/day) Islam, et al. [46];

CEa = Mean concentrations of elements observed in rice (µg/kg), for As, inorganic As (µg/kg) was considered; and

BW = Body weight (60 kg);

### 2.6. Health Risk Assessment

The calculation procedures for health risk assessment based on the hazard quotient (HQ) [48] and the incremental lifetime cancer risk (ILCR) from ingestion of TEs using US EPA criteria are described in Appendix A and Equations (1) and (2).
(1)$$\mathrm{HQ}=\frac{\mathrm{EDI}}{\mathrm{Rfd}}$$
ILCR = EDI × SF(2)

Detailed procedures were reported in our previous publication [41].

### 2.7. Quality Control and Assurance

Ultrapure water (ELGA PureLab classic, Woodridge, IL, USA) with 18 MΩ·cm resistivity was used for the preparation of the solution. Standard Reference Material (SRM) such as rice flour (SRM1568b) from the NIST (National Institute of Standard and Technology, Gaithersburg, MD, USA) was used to verify the analytical results of trace elements and nutrient elements in the rice samples. The analytical results of the SRM samples were checked after following the same technique of digestion and analysis as that of the samples. During analysis of TEs and nutrient elements by the ICP-MS, reagent blanks, duplicate samples, and spikes were also integrated. The experimental values were in good agreement with the certified values (Appendix A).

### 2.8. Statistical Analysis

JMP 14 pro software and Microsoft Excel 2016 were used for the statistical analysis of the data. Concentrations of trace elements and nutrient elements were reported as mean values with standard deviation (SD). Comparison between means was performed using the Student *t*-test. One or two-way analyses of variance were accomplished after checking the homogeneity of variance and normality of the data to assess the influence of washing and cooking procedures with the significance threshold set to *p* values less than 0.05.

## 3. Results and Discussion

### 3.1. Concentrations of TEs in Raw Rice of Different Varieties

Concentrations of As, Cd, and Pb in raw rice are presented in Figure 1a. Concentration differences of As, Cd, and Pb among the tested rice brands were nonsignificant. The highest raw rice Cd concentration was found in the BR 22 (167.3 µg/kg) variety, whereas the lowest Cd concentration was found in BRRIdhan 50 (11.6 µg/kg). Previously we found that the mean Cd concentration in rice grains collected from different districts of Bangladesh was 44 µg/kg (range 1 to 180 µg/kg) [28], which was similar to the findings of this study. A study from Bangladesh reported Cd concentration in raw rice was 33.1 ± 8.5 µg/kg, which was significantly lower than our findings [49]. Another study from China reported Cd content in raw rice collected from a market was 117 µg/kg [50]. Sharafi, et al. [51] found Cd content in Indian, Pakistani, and Iranian raw rice in the range of 19.9 to 146.2 µg/kg. Naseri, et al. [52] reported the Cd level in Iranian raw rice was 330 µg/kg, which was much higher than past studies.

Arsenic content in raw rice was the highest in BRRIdhan 50 (287.5 µg/kg) and lowest in BRRIdhan 34 (65.4 µg/kg). Khan, et al. [49] reported the average As content of raw rice was 240.8 µg/kg (range 219–253 µg/kg), which was similar to our findings. Ohno, et al. [53] reported the raw rice As concentration in Bangladeshi rice was 340 µg/kg. Another study from West Bengal, India reported As concentration in 15 raw rice samples varied between 111 and 540 µg/kg [21], which was higher than our findings. Zhuang, et al. [50] reported As content in raw rice collected from a Chinese market was 142 µg/kg. Sharafi, et al. [51] reported As concentration in Indian, Pakistani, and Iranian raw rice varied between 84.6 and 172.6 µg/kg. Mwale, et al. [18] reported average As content in UK, Sri Lanka, Myanmar, and Nigeria’s raw rice was 132 µg/kg. They revealed that As concentrations in rice varied based on geographical locations and rice types.

The study showed lower concentrations of Pb in all examined rice; the highest concentration was found in BR 22 (25.3 µg/kg) and the lowest content was found in BRRIdhan 34 (7 µg/kg). Sharafi, et al. [51] reported Pb content in raw rice varied from 47.5 to 1208.4 µg/kg, which was much higher than our findings. Behrouzi, et al. [54] reported mean Pb content in Iranian raw rice was 87 ± 4.8 µg/kg. Another study from Iran reported the raw rice concentration of Pb was 1750 µg/kg [52]. Batista [55] reported Pb levels in Brazilian rice varied between 0.4 and 14.5 µg/kg.

### 3.2. Effect of Different Washing Techniques on Reduction in TEs from Rice

The reduction percentage of As, Cd, and Pb from raw rice due to different washing procedures is presented in Figure 1b–d. For all three TEs, a higher reduction was found from washing five times compared to three of raw rice with water, though these variations were found to be nonsignificant. The average percentage reduction from washing of different brands of rice three and five times was 1.5 and 2 for Cd, 1.6 and 2.2 for As, and 5.3 and 6.3 for Pb, respectively. Previously, a study from Bangladesh reported 13–15% removal of As in raw rice when rice was washed until the water became clear [49]. Sengupta, et al. [21] reported 23% removal of As from raw rice due to washing five to six times with low As containing water. Al-Saleh and Abduljabbar [56] reported a 65% reduction in Cd in raw rice when grains were rinsed three times with deionized water. Behrouzi, et al. [54] reported a 37% reduction in Pb from Iranian raw rice when washed six times with deionized water. Many studies have demonstrated that washing of rice prior to cooking can efficiently reduce the levels of trace elements such as As, Cd, and Pb in cooked rice samples, which is in agreement with the results of this study [49,57,58,59,60]; although, the higher percentage of reduction in TEs from washing in other studies may be due to the soaking of rice with water for longer period of time and using a higher volume of water in washing, which could influence a higher reduction in TEs.

### 3.3. As Speciation and Reduction in Inorganic As from Rice

Speciation analysis of As was conducted on all (raw and cooked) rice samples. The results of As speciation in different brands of rice are presented in Table 1. The concentration of inorganic As (sum of arsenite and arsenate) in raw rice ranged between 52 and 238 µg/kg, whereas in cooked rice, inorganic As concentrations varied from 31 to 177 µg/kg. The overall average extraction efficiency by 1% nitric acid from different brands of rice was 84.7% of the total As, determined by nitric acid digestion. Past research showed that 59 to 98% As was found in rice as inorganic As [46]. Various studies on rice from the Bengal delta regions showed that inorganic forms of As were primarily present [61,62].

The influence of different washing and cooking methods on the percent reduction in inorganic As from raw rice is presented in Table 2. We found 1–17% reduction in inorganic As from raw rice when cooked in a rice cooker with a 1:2 rice to water ratio after washing three times, while a 20–39% reduction was observed in traditional cooking using a 1:6 rice to water ratio after washing five times. Atiaga, et al. [38] reported a 1:6 rice to water ratio reduced inorganic As by 80% from brown and 60% from white rice, while [25] reported a 44 and 49% reduction in inorganic As from brown and white rice, respectively, when cooked with excess water. Similarly Gray, et al. [26] reported a reduction in inorganic As by 40 and 50% from white and brown rice in cooking with 1:6–10 rice to water ratios. We can say that when rice was cooked with excess water (1:6), the percent reduction in inorganic As was higher compared to a 1:2 rice:water ratio.

### 3.4. Effect of Various Cooking Procedures on the Release of TEs from Rice

The average reduction in TEs from different brands of rice in various washing and cooking procedures are presented in Table 3. For As, Cd, and Pb, the lowest reduction was found when the rice grains were cooked by the rice cooker method using 1:2 rice to water ratio after washing three times, and the highest reduction was observed in traditional cooking with 1:6 rice to water ratio after washing rice grains five times. The percentage reduction in As from different rice brands due to different cooking procedures is presented in Figure 2a. We found 33% (range 26–42%) average reduction in As from tested rice brands when the rice grains were cooked in traditional methods with 1:6 rice to water ratio after washing with water five times. On the other hand, only a 7% reduction in rice grain As concentration was observed when the rice grains were cooked in a rice cooker with a 1:2 rice to water ratio after washing with water three times. Sengupta, et al. [21] reported 57% removal of As from rice when cooked with rice:water 1:6 after washing five to six times with water, which was higher than the findings of our study. Khan, et al. [49] reported a 54–57% reduction in As from Bangladeshi rice when cooked with an excessive volume of water after washing until water was clear, which was similar to the report by [21] but higher than our report. Zhuang, et al. [50] reported 3.5–6% reduction in As from raw rice when cooked with a Chinese traditional method (rice: water 1:2), which was very similar to our findings (7%). Sharafi, et al. [51] reported 25.8% removal of As when cooked with 1:2 rice:water ratio, while 44.2% of As was removed from raw rice when cooked with 1:4 rice:water ratio. Mwale, et al. [18] reported a 4.5% and 30% reduction in total As from raw rice when cooked with 1:3 and 1:6 rice to water ratios, respectively. Cooking of rice with 1:6 rice to water ratio was reported to remove As by 35% [25], 15 to 50% [26], and up to 63% [63]. Cooked rice As levels may vary from uncooked rice based on the As content in the cooking water [53,64,65].

The influence of different cooking procedures and washing on Cd removal percentage from various brands of rice is shown in Figure 2b. From our study, the average reduction in Cd from tested rice grains was found to be 35% (range 24–46%) when cooked by the traditional method with a 1:6 rice to water ratio and washing the raw rice with water five times. On the contrary, a 9% reduction in Cd in cooked rice was observed when rice grains were cooked in a rice cooker with a 1:2 rice to water ratio after washing with water three times. Zhuang, et al. [50] reported a 10% decrease in Cd from raw rice when cooked with a rice:water ratio 1:2, which was similar to our findings (9%). Sharafi, et al. [51] reported a 20.3% reduction in Cd when cooked with 1:2 rice:water ratio but a 26.9% reduction when cooked with 1:4 rice:water ratio. Surprisingly, Khan, et al. [49] found no reduction in Cd in cooked rice when cooked with an excessive amount of water.

The effect of various cooking methods on the percentage of reduction in Pb from different rice brands is presented in Figure 2c. The average reduction in Pb from different brands of rice was found to be 27% (range 19–32%) when cooked using traditional methods with 1:6 rice to water ratio after washing the raw rice with water five times, whereas an 8% reduction in Pb from raw rice was found when the rice grains were cooked in a rice cooker with a 1:2 rice to water ratio after washing the rice with water three times. Sharafi, et al. [51] reported a 26.9% reduction in Pb when cooked with a 1:2 rice:water ratio and a 44.3% reduction when cooked with a 1:4 rice:water ratio, which was higher than the findings from this study. A nonsignificant decrease in Pb concentration in cooked rice compared to raw rice was reported by Naseri, et al. [52] when cooked with a 1:7 rice to water ratio. A similar kind of result of the Pb concentration in cooked rice was demonstrated by Zazouli, et al. [66].

### 3.5. Influence of Cooking Procedures on Release of Essential Nutrients in Cooked Rice

Different brands of rice showed a significant loss of various essential nutrient elements when cooked by different procedures (Figure 3). With the increase in the volume of cooking water, a decrease in concentrations of nutrient elements was observed. The maximum average loss was observed for Se (49%) among all the essential nutrient elements when cooked with traditional methods with a 1: 6 rice to water ratio after washing five times with water. The concentration of Se varied between 42 and 314 µg/kg in raw rice of different brands used in this study. The highest concentration of Se was found in Banshful, whereas the lowest concentration was found from Binni rice. Copper was found least affected (10%) by different cooking procedures among all the essential elements, which was similar to the study of Mwale, et al. [18]. The raw rice concentration of Cu ranged from 1310 to 3608 µg/kg among different brands of rice, where the highest concentration was found in BRRIdhan 50 and the lowest in the BRRIdhan 34 variety of rice. We found the highest average reduction in other essential nutrients (Zn = 17%, Mn = 22%, and Mo = 22%) from raw rice, when rice grains of different brands were cooked with a 1: 6 rice to water ratio by the traditional method after washing with water five times. Mwale, et al. [18] also reported a significant reduction in essential elements (Zn = 7.7%, Cu = 0.2%, Mn = 16.5%, Se = 12%, and Mo = 38.5%) when rice was cooked in a 1:6 rice to water ratio. Menon, et al. [24] reported no significant reduction in Zn and Se but a 5% reduction in Mn from white and brown rice in parboiled and absorbed cooking treatments. Similarly, a reduction in essential elements from raw rice when cooked with excess water was reported by Pogoson, et al. [67], Carey, et al. [39] and Sharafi, et al. [1].

### 3.6. Daily Consumption of TEs and Health Risk

The estimated daily intake of As, Cd, and Pb from consumption of the tested brands of rice when cooked in a rice cooker with a 1:2 rice to water ratio after washing the rice three times with water and the associated health risks are presented in Table 4, while Table 5 represents the daily intake of TEs and associated health risk from rice consumption when cooked by the traditional methods after washing the rice grains with water five times, followed by cooking with a 1:6 rice to water ratio. For all the tested brands of rice, a higher daily intake of TEs and health risks (HQ and ILCR) were observed when rice grains were cooked by the rice cooker method with a low rice to water ratio (1:2) compared to the traditional method with excessive water (1:6 rice to water). Daily intake rates were found between 0.36 and 1.67 µg/kg bw for As, 0.06 and 1.15 µg/kg bw for Cd, and 0.04 and 0.17 µg/kg bw for Pb when rice grains were cooked by the rice cooker method, whereas in the traditional method, daily intake rates were ranged from 0.23 to 1.3 µg/kg bw for As, 0.04 to 0.88 µg/kg bw for Cd, and 0.03 to 0.15 µg/kg bw for Pb. Daily intake of As from this study (both rice cooker and traditional methods of cooking) was found lower than the JECFA benchmark value for daily As intake (2–7 µg/kg bw) (WHO 2011). Rahman et al. [17] found that adult males and females consumed 0.82 and 0.78 µg/kg bw inorganic As per day from rice in Bangladesh. Daily intake of As from Bangladeshi cooked rice was found to be 2.66 to 4.33 µg/kg bw [68].

Another study from Bangladesh reported a daily As intake from cooked rice to be 1.5 µg/kg bw [53]. The estimated daily intake of total As for young American males is 0.64 µg/kg bw, whereas, for Canadian males of a similar age group, the daily As intake rate is 0.98 µg/kg bw, and much higher values (2.66 to 4.66 µg/kg bw) were reported for Japanese females [69,70]. Jung, et al. [71] reported the daily intake of As from South Korean rice was 0.52 µg/kg bw.

Exposure to As may result in diabetes, hypertension, cardiovascular diseases, nervous and respiratory systems disorders, and even cancers in the liver, kidney, lung, bladder and skin [72,73]. The HQ and ILCR values for As from rice consumption ranged from 1.2 to 5.59 and 5.4 × 10^−4^ to 2.5 × 10^−3^, respectively when rice grains were cooked by the rice cooker method (1:2 rice to water ratio) whereas, using the traditional cooking method (1:6 rice to water ratio), HQ and ILCR values for As from rice consumption varied between 0.76 and 4.32 and 3.4 × 10^−4^ and 1.9 × 10^−3^, respectively. All the tested rice brands except BRRIdhan 34 had HQ values more than 1, suggesting the probability of noncarcinogenic risk for As consumption from rice. An acceptable ILCR varying between 10^−4^ (a 1:10,000 chance of developing cancer in a lifetime of a person) and 10^−6^ (a 1:1,000,000 chance of developing cancer in a lifetime of a person) has been set by USEPA [48]. Our ILCRs for As for the rice samples (rice cooker and traditional methods of cooking) were higher than the USEPA acceptable limit of 1 × 10^−4^ or 1 ×10^−6^, which reveals that consumption of As-contaminated rice brands poses risk of cancer to the consumers of Bangladesh.

The weekly tolerable Cd intake limits of ATSDR, EFSA, and JECFA are 0.7, 2.5, and 5.8 µg/kg bw, respectively [74,75,76]. In our study, we found the average weekly intake of Cd from consumption of different brands of rice was 3.03 µg/kg by the rice cooker method, which was higher than ATSDR and EFSA but lower than the JECFA recommendation, and 2.24 µg/kg by the traditional method, which was higher than the PTWI set by ATSDR but lower than the recommendation set by EFSA and JECFA. Al-Saleh and Abduljabbar [56] reported the mean weekly Cd from rice sold in Saudi Arabia was 0.503 µg/kg bw, which was lower than the findings of this study. Naseri, et al. [52] reported the average weekly intake of Cd from Iranian rice was 4.286 µg/kg bw, which was higher compared to the report of this study. In another study Naseri, et al. [77] reported a weekly intake of Cd from domestic and imported rice in Iran varied between 3.465 and 5.762 µg/kg bw which was also higher than the result of this study. Jung, et al. [71] reported a daily Cd intake from rice in South Korea of 5.38 µg, which was lower than the average daily intake of Cd (19.26 µg) found in this study from consumption of different brands of rice. The HQ and ILCR values for Cd from rice consumption ranged from 0.06 to 1.15 and 2.4 × 10^−5^ to 4.3 × 10^−4^, respectively, when rice grains were cooked by the rice cooker method (1:2 rice to water ratio), whereas, using the traditional cooking method (1:6 rice to water ratio) HQ and ILCR values for Cd from rice consumption varied between 0.04 and 0.88 and 1.6 × 10^−5^ and 3.3 × 10^−4^, respectively. All the tested rice brands except BR 22 had HQ values less than 1, suggesting no appreciable noncarcinogenic risk for Cd consumption from rice. The ILCR values for Cd for all the tested rice samples (rice cooker and traditional methods of cooking) were within the USEPA acceptable limit of 1 × 10^−4^ or 1 ×10^−6^ suggesting that consumption of Cd-contaminated rice brands poses no appreciable carcinogenic risk to the consumers of Bangladesh.

Three different benchmark doses were set by the EFSA for daily dietary intake of Pb, which are: 0.5 µg/kg bw (for neurodevelopmental effects), 0.63 µg/kg bw (for nephrotoxic effects), and 1.5 µg/kg bw (for cardiovascular effects) (EFSA 2012). In our study, we found the average weekly intake of Pb from consumption of different brands of rice was 0.59 µg/kg bw using the rice cooker method and 0.48 µg/kg bw using the traditional method, which were lower than the EFSA benchmark values. Al-Saleh and Abduljabbar [56] reported the weekly intake of Pb from Saudi Arabian rice was 1.068 µg/kg bw, which was higher than the report of this study. Naseri, et al. [52] found the weekly Pb intake from Iranian rice was 22.458 µg/kg bw, which was very high compared to the findings of this study. Naseri, et al. [77] also reported a weekly intake of Pb from domestic and imported rice in Iran from 9.189 to 25.769 µg/kg bw. Daily intake of Pb from rice consumption in Brazil was found to be 0.44 µg [55], which was similar to the result the of rice cooker method (0.43 µg/kg bw) but higher than the traditional method (0.32 µg/kg bw). In South Korea, daily intake of Pb from rice was estimated to be 52.7 µg [71].

Exposure to Pb at a lower level has been linked with cardiovascular, neurological, immunological, renal, reproductive, and developmental effects [56]. The HQ and ILCR values for Pb from rice consumption ranged from 0.003 to 0.01 and 3.8 × 10^−7^ to 1.4 × 10^−6^, respectively, when rice grains were cooked by the rice cooker method (1:2 rice to water ratio), whereas, using the traditional cooking method (1:6 rice to water ratio), the HQ and ILCR values for Pb from rice consumption varied between 0.003 and 0.01 and 2.9 × 10^−7^ and 1.2 × 10^−6^, respectively. All the tested rice brands had HQ values less than 1 and ILCR values within the permissible level suggesting no appreciable noncarcinogenic and carcinogenic risk for Pb consumption from rice.

## 4. Conclusions

In this study, the influence of washing and cooking procedures on the reduction in TEs (As, Cd, and Pb), including inorganic As and essential nutrient elements were evaluated. It is evident from this study that washing rice five times prior to cooking can remove slightly more TEs than washing three times. Out of four different rice to water ratios (1:2, 1:3, 1:4, and 1:6) used for cooking, it was found that the concentration of TEs in rice was effectively reduced through cooking rice in excess water (1:6 rice to water ratio), but the loss of essential nutrient elements in this procedure is a concern. Both noncarcinogenic and carcinogenic risks for Cd and Pb via rice consumption were negligible. However, for As, the values showed higher risks than the acceptable values, even after the reduction by increasing the water ratio for rice cooking along with washing the rice with water five times.

## Figures and Tables

**Figure 1 ijerph-19-02582-f001:**
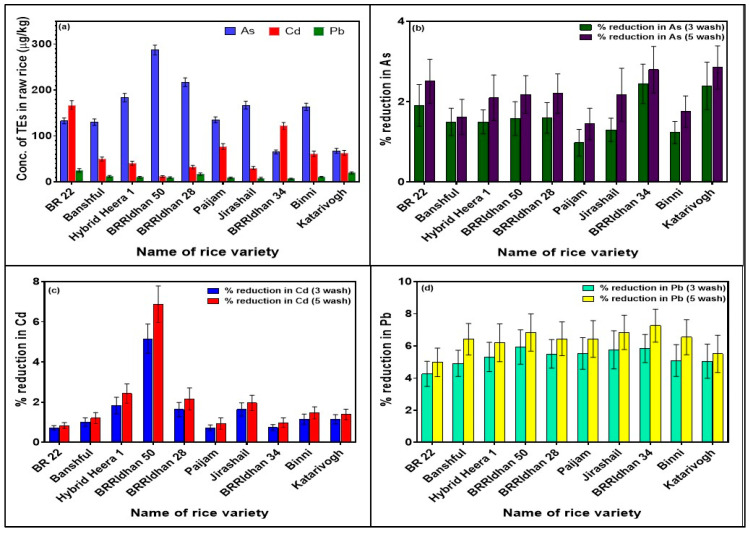
(**a**) Concentrations of TEs (As, Cd, and Pb) in raw rice; and effect of various washing procedures on percentage of reduction in TEs: (**b**) As, (**c**) Cd, and (**d**) Pb from raw rice.

**Figure 2 ijerph-19-02582-f002:**
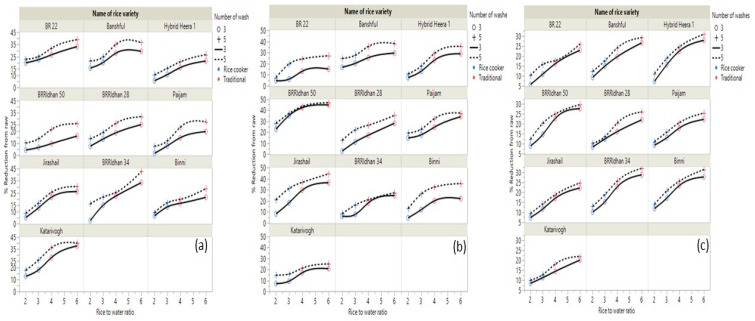
Effect of various cooking methods on percentage reduction in (**a**) As, (**b**) Cd, and (**c**) Pb from different rice brands.

**Figure 3 ijerph-19-02582-f003:**
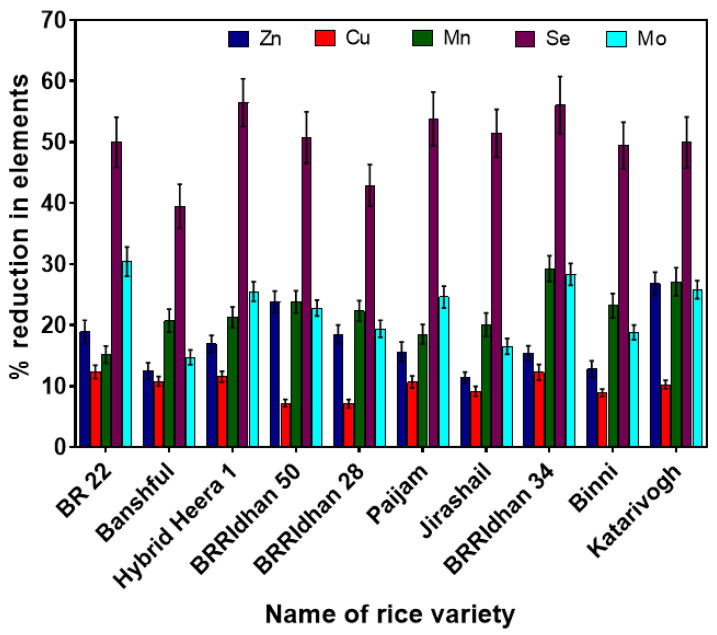
Influence of various cooking procedures on removal of different essential nutrient elements (Zn, Cu, Mn, Se, and Mo) from raw rice.

**Table 1 ijerph-19-02582-t001:** Mean total and inorganic As concentration (µg/kg) in different brands of rice with recovery percentage.

Name of Rice Brands	Total As in Raw Rice	Inorganic As in Raw Rice	Total As in Cooked Rice	Inorganic As in Cooked Rice	% of Inorganic As in Raw Rice	% of Inorganic As in Cooked Rice
BR 22	133 ± 6	109 ± 12	81 ± 5	69 ± 9	82	85
Banshful	130 ± 7	105 ± 10	82 ± 7	67 ± 7	81	81
Hybrid Heera 1	183 ± 9	157 ± 18	134 ± 8	115 ± 13	85	86
BRRIdhan 50	287 ± 11	238 ± 17	212 ± 15	177 ± 14	82	83
BRRIdhan 28	217 ± 10	180 ± 16	148 ± 9	132 ± 14	82	89
Paijam	135 ± 6	114 ± 9	98 ± 4	84 ± 8	84	86
Jirashail	167 ± 8	138 ± 11	116 ± 7	100 ± 9	82	85
BRRIdhan 34	65 ± 4	52 ± 7	37 ± 3	31 ± 4	80	84
Binni	163 ± 8	132 ± 9	116 ± 6	115 ± 10	81	84
Katarivogh	68 ± 5	56 ± 5	40 ± 4	34 ± 4	83	84

**Table 2 ijerph-19-02582-t002:** Percent reduction in inorganic As from raw rice due to different washing and cooking methods.

Rice Brands	Inorganic As in Raw Rice (µg/kg)	Percent Reduction in Inorganic As from Raw Rice							
		Washing 3 Times				Washing 5 Times			
		Rice Cooker		Traditional		Rice Cooker		Traditional	
		1:2	1:3	1:4	1:6	1:2	1:3	1:4	1:6
BR 22	109 ± 12	17 ± 3	23 ± 5	25 ± 4	29 ± 6	20 ± 3	21 ± 3	27 ± 4	36 ± 5
Bashful	105 ± 10	14 ± 2	18 ± 4	27 ± 5	28 ± 4	23 ± 2	26 ± 4	32 ± 4	36 ± 4
Hybrid Heera 1	157 ± 18	6 ± 2	12 ± 3	18 ± 3	23 ± 2	9 ± 2	19 ± 3	23 ± 2	26 ± 3
BRRIdhan 50	238 ± 17	2 ± 1	6 ± 2	10 ± 2	11 ± 2	6 ± 1	8 ± 1	16 ± 3	25 ± 3
BRRIdhan 28	180 ± 16	6 ± 1	12 ± 3	18 ± 2	23 ± 3	9 ± 1	16 ± 2	25 ± 3	27 ± 2
Paijam	114 ± 9	1 ± 0.3	5 ± 2	10 ± 3	16 ± 2	5 ± 0.5	14 ± 2	18 ± 2	25 ± 3
Jirashail	138 ± 11	3 ± 1	12 ± 3	18 ± 3	26 ± 4	5 ± 1	15 ± 3	20 ± 3	27 ± 4
BRRIdhan 34	52 ± 7	2 ± 0.5	9 ± 2	19 ± 4	28 ± 4	12 ± 2	14 ± 2	19 ± 2	37 ± 5
Binni	132 ± 9	1 ± 0.4	2 ± 1	6 ± 1	13 ± 2	6 ± 1	8 ± 1	11 ± 2	20 ± 3
Katarivogh	56 ± 5	11 ± 3	16 ± 3	24 ± 4	34 ± 5	15 ± 3	22 ± 3	32 ± 3	39 ± 4

**Table 3 ijerph-19-02582-t003:** Average reduction (%) in TEs from different brands of rice in different cooking procedures.

Washing Procedures	Cooking Methods	Rice to Water Ratio	Percent Reduction in TEs from Raw Rice		
			As	Cd	Pb
3 times	Rice cooker	1:2	7.5 ± 3.5	9.5 ± 4.3	8.6 ± 1.7
		1:3	13 ± 6.5	15 ± 8.4	13.5 ± 2.5
	Traditional	1:4	19 ± 8	23 ± 7.3	18.7 ± 4.0
		1:6	25 ± 8.7	28.5 ± 8.5	24 ± 4.2
5 times	Rice cooker	1:2	13 ± 6.4	16 ± 6.9	11 ± 1.7
		1:3	18 ± 6.7	23.7 ± 6.8	16 ± 3.5
	Traditional	1:4	25.5 ± 8.2	30 ± 7.2	21.6 ± 3.9
		1:6	33 ± 6.1	35 ± 8.1	27 ± 4.3

**Table 4 ijerph-19-02582-t004:** Estimated daily intake of TEs (As, Cd, and Pb) from different brands of rice and the health risk for washing 3 times and a rice to water ratio of 1:2 (rice cooker methods of cooking).

Name of Rice Variety	Mean Conc. of TEs (µg/kg)			Average Daily Intake of Rice (kg/Day)	Daily Intake of TEs (µg/kg bw)			HQ			ILCR		
	As	Cd	Pb		As	Cd	Pb	As	Cd	Pb	As	Cd	Pb
BR 22	90.0 ± 15	159.4 ± 24	23.9 ± 4.4	0.432	0.65	1.15	0.17	2.16	1.15	0.01	9.7 × 10^−4^	4.3 × 10^−4^	1.4 × 10^−6^
Banshful	92.8 ± 12	41.3 ± 4	10.9 ± 3.2		0.67	0.3	0.08	2.23	0.3	0.006	1.0 × 10^−3^	1.1 × 10^−4^	6.6 × 10^−7^
Hybrid Heera 1	146.8 ± 19	37.3 ± 5	9.4 ± 1.2		1.06	0.27	0.07	3.52	0.27	0.005	1.5 × 10^−3^	1.0 × 10^−4^	5.7 × 10^−7^
BRRIdhan 50	232.9 ± 25	9.0 ± 1	9.0 ± 1.7		1.67	0.06	0.06	5.59	0.06	0.005	2.5 × 10^−3^	2.4 × 10^−5^	5.5 × 10^−7^
BRRIdhan 28	170.1 ± 15	31.4 ± 5	15.6 ± 2.9		1.22	0.23	0.11	4.08	0.23	0.009	1.8 × 10^−3^	8.6 × 10^−5^	9.5 × 10^−7^
Paijam	112.8 ± 12	65.6 ± 4	8.0 ± 1.3		0.81	0.47	0.06	2.70	0.47	0.005	1.2 × 10^−3^	1.8 × 10^−4^	4.9 × 10^−7^
Jirashail	134.6 ± 10	28.0 ± 2	7.7 ± 1.1		0.97	0.2	0.05	3.23	0.2	0.004	1.4 × 10^−3^	7.6 × 10^−5^	4.7 × 10^−7^
BRRIdhan 34	53.9 ± 3	114.3 ± 10	6.3 ± 0.9		0.39	0.82	0.04	1.29	0.82	0.003	5.8 × 10^−4^	3.1 × 10^−4^	3.8 × 10^−7^
Binni	137.9 ± 13	58.3 ± 6	9.7 ± 1.2		1.0	0.42	0.06	3.31	0.42	0.006	1.4 × 10^−3^	1.6 × 10^−4^	5.9 × 10^−7^
Katarivogh	50.3 ± 5	58.8 ± 7	18.1 ± 3.1		0.36	0.42	0.13	1.20	0.42	0.01	5.4 × 10^−4^	1.6 × 10^−4^	1.1 × 10^−6^

**Table 5 ijerph-19-02582-t005:** Estimated daily intake of TEs (As, Cd, and Pb) from different brands of rice and the health risk for washing 5 times and a rice to water ratio of 1:6 (traditional methods of cooking).

Name of Rice Variety	Mean Conc. of TEs (µg/kg)			Average Daily Intake of Rice (kg/Day)	Daily Intake of TEs (µg/kg bw)			HQ			ILCR		
	As	Cd	Pb		As	Cd	Pb	As	Cd	Pb	As	Cd	Pb
BR 22	68.7 ± 10	121.9 ± 28	20.5 ± 3.4	0.432	0.49	0.88	0.15	1.65	0.88	0.01	7.4 × 10^−4^	3.3 × 10^−4^	1.2 × 10^−6^
Banshful	69.8 ± 17	30.7 ± 7	8.5 ± 1.5		0.5	022	0.06	1.68	0.22	0.005	7.5 × 10^−4^	8.4 × 10^−5^	5.2 × 10^−7^
Hybrid Heera 1	113.8 ± 8	26.0 ± 3	7.0 ± 1.0		0.82	0.19	0.05	2.73	0.19	0.004	1.2 × 10^−3^	7.1 × 10^−5^	4.3 × 10^−7^
BRRIdhan 50	179.9 ± 21	6.2 ± 0.9	6.9 ± 1.2		1.3	0.04	0.05	4.32	0.04	0.004	1.9 × 10^−3^	1.6 × 10^−5^	4.2 × 10^−7^
BRRIdhan 28	125.3 ± 18	20.9 ± 5	12.6 ± 2.7		0.9	0.15	0.09	3	0.15	0.007	1.3 × 10^−3^	5.7 × 10^−5^	7.7 × 10^−7^
Paijam	82.9 ± 10	48.1 ± 7	6.6 ± 1.6		0.6	0.35	0.05	1.99	0.35	0.004	8.9 × 10^−4^	1.3 × 10^−4^	4.0 × 10^−7^
Jirashail	98.6 ± 21	17.0 ± 1.2	6.3 ± 1.2		0.71	0.12	0.05	2.37	0.12	0.004	1.1 × 10^−3^	4.6 × 10^−5^	3.8 × 10^−7^
BRRIdhan 34	31.8 ± 9	88.5 ± 8	4.7 ± 1.3		0.23	0.64	0.03	0.76	0.64	0.003	3.4 × 10^−4^	2.4 × 10^−4^	2.9 × 10^−7^
Binni	98.6 ± 8	39.4 ± 2	7.5 ± 1.4		0.71	0.28	0.05	2.37	0.28	0.004	1.1 × 10^−3^	1.1 × 10^−4^	4.6 × 10^−7^
Katarivogh	34.5 ± 4	47.3 ± 6	15.4 ± 2.7		0.25	0.34	0.11	0.83	0.34	0.009	3.7 × 10^−4^	1.3 × 10^−4^	9.4 × 10^−7^

## Data Availability

Data will be made available on reasonable request.

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
