# Peer review of "Removal of Toxic and Essential Nutrient Elements from Commercial Rice Brands Using Different Washing and Cooking Practices: Human Health Risk Assessment"

_ijerph, 2022, doi:10.3390/ijerph19052582_

Round 1
Reviewer 1 Report
Very briefly, the introduction must be modified and the references here cited must be updated. Doubts on the overall reliability are present among the speciation methods used for As species, as well as on the Cd recovery using an open system of digestion. All methods assessed need to be completely validated in terms of sensitivity, linearity, precision and trueness. All remarks are reported as sticky notes in the pdf file enclosed.

Author Response
Comment: Very briefly, the introduction must be modified and the references here cited must be updated. Doubts on the overall reliability are present among the speciation methods used for As species, as well as on the Cd recovery using an open system of digestion. All methods assessed need to be completely validated in terms of sensitivity, linearity, precision and trueness.
Reply: We highly appreciate reviewer’s comments on our manuscript. As per reviewer’s suggestion, we have modified the introduction section with more focus on arsenic in rice instead of cadmium.
Reviewer has raised issues on the analytical methodologies used for arsenic speciation as well as open vessel digestion of rice samples for cadmium included in this study. It should be noted that all analytical procedures are well stablished and published in various high impacted journals by authors. However, we have cited all relevant references to avoid any confusions.
Comment: 1) The space dedicated in the introduction to the three toxic elements does not reflect their importance in the results obtained. For example, almost all of the highlighted part is related to cadmium, which was found non-critical in the rice examined. On the other hand, only one-line space was dedicated to lead and arsenic. In particular, the bioaccumulation of arsenic in rice has been a topic of worldwide resonance in the last thirty years, and many efforts have been made to reduce its bioaccumulation in the edible fraction, even in heavily polluted soils. On the other hand, the most significant results of this study were obtained precisely on the reduction of arsenic concentration. It is therefore necessary to completely rewrite this part, reducing the space dedicated to cadmium and increasing that dedicated to arsenic, and inserting the references of the most significant results obtained so far in an attempt to minimize their bioaccumulation in rice (e.g. 10.1021/es300636d; 10.1002/jsfa.3648; 10.1007/s10653-013-9533-z; 10.1016/j.scitotenv.2018.02.157; 10.1016/j.chemosphere.2021.130351).
Reply: We agree with the reviewer and have revised the Introduction section with more focusing on arsenic. However, the above-mentioned citations given by reviewers are for irrigation options (water management) of paddy rice and uptake of arsenic and cadmium, which are not relevant to our study as our study focussed on washing and cooking effects on arsenic, cadmium and lead retention/removal from cooked rice. However, we have cited all references in the revised manuscript mentioned by reviewers. Please see pages 2-3 and lines 37 – 53.
Comment: 2) It is necessary to report the concentration of As, Cd and Pb measured in tap water.
Reply: We have included the concentration of As, Cd and Pb in tap water in the revised manuscript as per reviewer’s suggestion. They are all below the instrument detection limits (line 105).
Comment: 3) Please perform the validation of the method used, in terms of LoD, LoQ, linearity and precision along the operative range of concentration.
Reply: Please note that all analytical methods were validated and reported in several published articles. However, we have included this information along with citations in the revised manuscript (lines 128-130).
Kumar, M., Rahman, M.M., Ramanathan, A.L. and Naidu, R., 2016. Arsenic and other elements in drinking water and dietary components from the middle Gangetic plain of Bihar, India: health risk index. Science of the Total Environment, 539, pp.125-134.
Mwale, T., Rahman, M.M. and Mondal, D., 2018. Risk and benefit of different cooking methods on essential elements and arsenic in rice. International journal of environmental research and public health, 15(6), p.1056.
Comment: 4) Cadmium is a very volatile element, and losses using an open heating block digestor are very frequent. The reliability of the whole procedure of sample digestion should be ensured only by analysis of a more than one certified reference materials, also at very different Cd concentration.
Reply: We thank reviewer for the comment. In addition to microwave digestion, open vessel digestion is very popular for analysing trace elements in various food samples including rice and vegetables. This digestion method has been well reported by our team (references below) and other scientists. Previously we have used different SRM (Montana soil, Rice and Tomato leaves) to validate this method. However, we have included 2 more SRMs (Tomato leaves and spinach leaves) in the revised manuscript. Please see Table S2 (Supplementary Information).
Shahriar, S., Rahman, M.M. and Naidu, R., 2020. Geographical variation of cadmium in commercial rice brands in Bangladesh: Human health risk assessment. Science of The Total Environment, 716, p.137049.
Shahriar, S., Haque, M.M., Naidu, R. and Rahman, M.M., 2021. Concentrations of toxic elements and health risk assessment in arum grown in arsenic-contaminated areas of Bangladesh. Food Control, 129, p.108240.
Rahman, M.M., Asaduzzaman, M. and Naidu, R., 2013. Consumption of arsenic and other elements from vegetables and drinking water from an arsenic-contaminated area of Bangladesh. Journal of hazardous materials, 262, pp.1056-1063.
Halder, D., Saha, J.K. and Biswas, A., 2020. Accumulation of essential and non-essential trace elements in rice grain: Possible health impacts on rice consumers in West Bengal, India. Science of The Total Environment, 706, p.135944.
Gawalko, E.J., Nowicki, T.W., Babb, J., Tkachuk, R. and Wu, S., 1997. Comparison of closed-vessel and focused open-vessel microwave dissolution for determination of cadmium, copper, lead, and selenium in wheat, wheat products, corn bran, and rice flour by transverse-heated graphite furnace atomic absorption spectrometry. Journal of AOAC International, 80(2), pp.379-387.
Simmons, R.W., Pongsakul, P., Saiyasitpanich, D. and Klinphoklap, S., 2005. Elevated levels of cadmium and zinc in paddy soils and elevated levels of cadmium in rice grain downstream of a zinc mineralized area in Thailand: implications for public health. Environmental geochemistry and health, 27(5-6), pp.501-511.
Comment: 5) Speciation of As is an analytical approach of extraordinary complexity, due the easiness of the possible inter-conversion among the As species. In particular, a number of literature findings accounted of a partial oxidatio of As(III) species to As(V) species using high concentrations of oxidizing acids like HNO3. Unfortunately, Authors seems to ignore all this. It is almost sure (see for instance 10.1016/j.jenvman.2021.114105; 10.1039/c002306; 10.1080/19440041003636661; j10.1016/j.talanta.2014.07.001) that - in such conditions - the relative amount of both inorganic As species is affected by a severe bias, but Authors have not realized this due to the absence of any form of validation of their analytical method. In addition, also the speciation method of As need to be completely validated!
Reply: We greatly appreciate reviewer’s comment. While acknowledging the partial oxidation of As(III) and As(V) using 1% HNO3 or 0.28M HNO3 or 2M TFA (trifluoracetic acid), we would like to inform that we have reported inorganic arsenic [sum of As(III) and As(V)] and hence did not report individual concentrations of As(III) and As(V) in cooked rice samples. Please be noted that this 1% HNO3 extraction method have already validated in literature (by other renowned researchers and authors team). We have cited some of these articles in the revised manuscript. Author’s team are highly experienced on arsenic speciation in rice and published vast numbers of articles. For reviewers’ convenience, please see below the references (lines 134-136).
Rahman, M.M., Alauddin, M., Alauddin, S.T., Siddique, A.B., Islam, M.R., Agosta, G., Mondal, D. and Naidu, R., 2021. Bioaccessibility and speciation of arsenic in children's diets and health risk assessment of an endemic area in Bangladesh. Journal of Hazardous Materials, 403, p.124064.
Islam, S., Rahman, M.M., Rahman, M.A. and Naidu, R., 2017. Food Control, 82, pp.196-202.
Islam, S., Rahman, M.M., Islam, M.R. and Naidu, R., 2017. Geographical variation and age-related dietary exposure to arsenic in rice from Bangladesh. Science of the Total Environment, 601, pp.122-131.
Shahriar, S., Haque, M.M., Naidu, R. and Rahman, M.M., 2021. Concentrations of toxic elements and health risk assessment in arum grown in arsenic-contaminated areas of Bangladesh. Food Control, 129, p.108240.
Rahman, M.M., Asaduzzaman, M. and Naidu, R., 2011. Arsenic exposure from rice and water sources in the Noakhali district of Bangladesh. Water Quality, Exposure and Health, 3(1), pp.1-10.
Rahman, M.A., Rahman, M.M., Reichman, S.M., Lim, R.P. and Naidu, R., 2014. Arsenic speciation in Australian-grown and imported rice on sale in Australia: implications for human health risk. Journal of agricultural and food chemistry, 62(25), pp.6016-6024.
Signes-Pastor, A.J., Carey, M. and Meharg, A.A., 2016. Inorganic arsenic in rice-based products for infants and young children. Food chemistry, 191, pp.128-134.
Rahman, M.M., Chen, Z. and Naidu, R., 2009. Extraction of arsenic species in soils using microwave-assisted extraction detected by ion chromatography coupled to inductively coupled plasma mass spectrometry. Environmental geochemistry and health, 31(1), pp.93-102.
Reviewer 2 Report
The authors studied the use of different washing and cooking methods to remove toxic and essential nutrients from commercial rice brands: human health risk assessment.Some meaningful conclusions are drawn, which are suggested to be revised and published.
- Whether the experimental design and data sources are further supplemented
- Reduce sentences appropriately
- How to avoid handling uncertainties in sample collection?

Author Response
Reviewer #2:
Comment: The authors studied the use of different washing and cooking methods to remove toxic and essential nutrients from commercial rice brands: human health risk assessment. Some meaningful conclusions are drawn, which are suggested to be revised and published.
Reply: We greatly appreciate reviewer for the excellent comment.
Comment: 1) Whether the experimental design and data sources are further supplemented
Reply: We have elaborated the experimental section in the revised manuscript.
Comment: 2) Reduce sentences appropriately
Reply: We are unable to understand this comment. No indication has been provided in the supplied pdf.
Comment: 3) How to avoid handling uncertainties in sample collection?
Reply: We thank the reviewer for the comment. We would like to inform that each experiment was repeated 3 times and samples were collected separately with clean gloves and stored them in separate zip-lock bags to avoid any cross contamination.
Reviewer 3 Report
Manuscript Number: ijerph-1538948
The manuscript (MS) titled "Removal of Toxic and Essential Nutrient Elements from Commercial Rice Brands Using Different Washing and Cooking Practices: Human Health Risk Assessment" has significant environmental health implications. However, However, this is a simple test with numerous experimental design issues. The MS was well structured and organized, but problems occurred in almost every section of the MS, particularly in "Materials and Methods," thus the "Results and discussion." More experiments must be conducted, and the manuscript must be rewritten before it can be submitted for publication.
Materials and methods are too brief and not clear enough. This is one of the main sections of the MS, and the method used should be described in sufficient detail for others to repeat your work. The author did not specify how many replicates they performed for their experiment. Did you analyze metals from each wash individually, or did you pool all three or five washing times and analyze for metals?
Consequently, the results are presented in tables without SD/SEM, but it is strange that in all the figures, you showed the SD/SEM, and I can see that those are possibly not the actual SD/SEM but simply the error bars you get from the Chart Design (Excel).
Some additional remarks:
- Equation 1: the last part (EF*ED/AT) can be omitted.
- Line 148: There is no more RfD for Pb because there is no evidence of a non-harmful intake threshold (Codex Alimentarius Commission, 2018). As a result, the THQ for Pb should be calculated as follows EU guideline (2006).
- Line 153: Cd's SF is 1.5 (OEHHA), not 0.38.
- Lines 174-175: The graph clearly shows the distinct differences between rice brands, not as the authors stated. However, I think you cannot draw this conclusion because it appears that it is not possible to perform statistical analysis in case the experiment was not repeated.
- Figure 1b is hard to read, and it is preferable to divide it into two figures.
Author Response
Reviewer #3:
Comment: The manuscript (MS) titled "Removal of Toxic and Essential Nutrient Elements from Commercial Rice Brands Using Different Washing and Cooking Practices: Human Health Risk Assessment" has significant environmental health implications. However, this is a simple test with numerous experimental design issues. The MS was well structured and organized, but problems occurred in almost every section of the MS, particularly in "Materials and Methods," thus the "Results and discussion." More experiments must be conducted, and the manuscript must be rewritten before it can be submitted for publication.
Reply: We have considered all comments raised by reviewer and revised the text accordingly.
Comment: 1) Materials and methods are too brief and not clear enough. This is one of the main sections of the MS, and the method used should be described in sufficient detail for others to repeat your work. The author did not specify how many replicates they performed for their experiment. Did you analyze metals from each wash individually, or did you pool all three or five washing times and analyze for metals?
Reply: We have elaborated the Materials and methods. We have used three replicates. We have analyzed metals from all three or five washing times together. This has been now clarified in the revised manuscript.
Comment: 2) Consequently, the results are presented in tables without SD/SEM, but it is strange that in all the figures, you showed the SD/SEM, and I can see that those are possibly not the actual SD/SEM but simply the error bars you get from the Chart Design (Excel).
Reply: We have included the SD in the revised Tables. It was our mistake.
Comment: 3) Equation 1: the last part (EF*ED/AT) can be omitted.
Reply: Done.
Comment: 4) Line 148: There is no more RfD for Pb because there is no evidence of a non-harmful intake threshold (Codex Alimentarius Commission, 2018). As a result, the THQ for Pb should be calculated as follows EU guideline (2006).
Reply: Agree. However, we have used USFDA 2018 value of 12.5 for Pb and revised the text accordingly.
Comment: 5) Line 153: Cd's SF is 1.5 (OEHHA), not 0.38.
Reply: Please note that Cd's SF is 1.5 (OEHHA) for inhalation and hence we have used 0.38 with appropriate reference.
Comment: 6) Lines 174-175: The graph clearly shows the distinct differences between rice brands, not as the authors stated. However, I think you cannot draw this conclusion because it appears that it is not possible to perform statistical analysis in case the experiment was not repeated.
Reply: We have used three replicates for each individual experiment and hence the conclusion is correct.
Comment: 7) Figure 1b is hard to read, and it is preferable to divide it into two figures.
Reply: We have divided the Figure 1b into 3 separate figures (1b-d) for better understanding.
Round 2
Reviewer 1 Report
The scientific level of the paper has been greatly improved. The paper can be published in its current form